# The Impact of Signing Do-Not-Resuscitate Orders on the Use of Non-Beneficial Life-Sustaining Treatments for Intensive Care Unit Patients: A Retrospective Study

**DOI:** 10.3390/ijerph19159521

**Published:** 2022-08-03

**Authors:** Shang-Sin Shiu, Ting-Ting Lee, Ming-Chen Yeh, Yu-Chi Chen, Shu-He Huang

**Affiliations:** 1Department of Nursing, Taipei Veterans General Hospital, Taipei 112201, Taiwan; ssshiu2@vghtpe.gov.tw; 2Department of Nursing, College of Nursing, National Yang Ming Chiao Tung University, Taipei 112304, Taiwan; tingting@nycu.edu.tw; 3Department of Nursing, Hungkuang University, Taichung 433304, Taiwan; mcyeh@sunrise.hk.edu.tw; 4Institute of Clinical Nursing, College of Nursing, National Yang Ming Chiao Tung University, Taipei 112304, Taiwan; ycchen2@nycu.edu.tw

**Keywords:** DNR, non-beneficial, resuscitation, withdrawal, life-sustaining treatments

## Abstract

Background: Intensive care medical technology increases the survival rate of critically ill patients. However, life-sustaining treatments also increase the probability of non-beneficial medical treatments given to patients at the end of life. Objective: This study aimed to analyse whether patients with a do-not-resuscitate (DNR) order were more likely to be subject to the withholding of cardiac resuscitation and withdrawal of life-sustaining treatment in the ICU. Methods: This retrospective study collected data regarding the demographics, illness conditions, and life-sustaining treatments of ICU patients who were last admitted to the ICU between 1 January 2016 and 31 December 2017, as determined by the hospital’s electronic medical dataset. Results: We identified and collected data on 386 patients over the two years; 319 (82.6%) signed a DNR before the end. The study found that DNR patients were less likely to receive cardiac resuscitation before death than non-DNR patients. The cardiac resuscitation treatments included chest compressions, electric shock, and cardiotonic drug injections (*p* < 0.001). However, the life-sustaining treatments were withdrawn for only a few patients before death. The study highlights that an early-documented DNR order is essential. However, it needs to be considered that promoting discussions of time-limited trials might be the solution to helping ICU terminal patients withdraw from non-beneficial life-sustaining treatments.

## 1. Introduction

The intensive care unit (ICU) is a unit in a hospital providing intensive care for critically ill patients with specially trained medical personnel and equipment that allows for continuous monitoring and life support. Significantly, most critically ill patients are sent to ICU to receive intensive care. They all require intensive medical care because of their severe condition; their physical condition is rapidly changing and they need life-sustaining treatments [1]. Consequently, the likelihood that health professionals will have to provide resuscitation practices of cardiopulmonary resuscitation (CPR) or various non-beneficial life-sustaining treatments to dying patients in the ICU increases if patients are without DNR orders. Cardiopulmonary resuscitation (CPR) is an emergency lifesaving procedure performed during a cardiac arrest [2]. The treatments include endotracheal intubation with a ventilator (ET), electric shocks, chest compressions, and cardiotonic drug injection to help the blood circulation, gas exchange, and ventilation to save patients at risk of death. In particular, most ICU patients are admitted after endotracheal intubation so that their ventilation care needs are met. Therefore, ET is a life-sustaining therapy for ICU patients.

In the ICU, trained physicians and nurses have to closely monitor the changes in the patients’ vital signs, observe the response to treatment, deal with disease-related changes quickly, prevent adverse effects, and provide physical and psychological comfort care to improve patients’ quality of life in the ICU. Medical staff and family members of patients hope that ICU care can improve the critical condition of patients to regain health. We can see that critical care science and technology have increased critically ill patient survival rates, but the current medical technology still has limits; it cannot prevent the death of an irreversibly severe or terminally ill patient. With patients with the above conditions, most nurses believe that despite great effort, life-sustaining treatments may prolong the lives and suffering of terminally ill patients [1], and DNR decisions are warranted to reduce the suffering of being administered ineffective emergency resuscitation practice [3,4]. 

However, many patients are unconscious and fail to discuss and make medical decisions and sign a Do-Not-Resuscitate (DNR) order regarding the refusal of resuscitation practices and non-beneficial life-sustaining treatments [5,6], or family members may not be able to complete an advanced consent DNR order for their patient because they cannot understand the actual illness situation, and may even act against the patient’s prior will [7]. Furthermore, sometimes DNR is misunderstood as a form of passive euthanasia [8,9], and euthanasia is still under debate with regard to clinical ethics and principles [4], and is incompatible with the healer role of the physician. Therefore, physicians might not actively discuss and complete DNR documentation. In fact, DNR is the solution for the clinical–ethical dilemma in end-of-life care, and can facilitate decision-making about withholding or withdrawing life-sustaining treatments or interventions [2,4]. DNR orders must be made with an emphasis on the basis of informed consent, advance directives, or surrogate decision [2].

To solve the above dilemma, in 2000, Taiwan enacted the Hospice Palliative Care Act legislation [10]. Under the doctor’s instructions, the terminally ill patient himself or his family can legally sign the DNR allowing the patient to refuse CPR and choose to withdraw from non-beneficial medical therapy. With the promotion of palliative care, public recognition of the non-performance of CPR on terminally ill patients has been increased to prevent them from receiving ineffective medical treatment. In 2010, a clinical research study found that more than 3/5 ICU patients signed DNR orders in the last 48 h of their lives, and the DNR order seemed to prevent some terminal patients from receiving resuscitation practices [11]. Despite the Hospice Palliative Care Act allowing withdrawal of ineffective treatments to reduce patient pain before death, “rescue to the end” is still a medical dilemma for critically ill ICU patients; that is, once the life-sustaining treatments are used, they will continue to be used until the patient dies. Even if the DNR is signed, the family may still request some seemingly less invasive medical treatments to prolong or save the patient’s life. This would prolong suffering in the end-stage of life for critically and terminally ill patients. Most ICU professionals have received palliative care education and recognise the importance of withholding resuscitation practices and withdrawing non-beneficial life-sustaining treatments; they would like to undertake palliative care to promote terminal patients’ comfort and facilitate natural death [3]. Medical professionals face clinical ethical dilemmas [12,13] and feel morally distressed and emotionally exhausted [14,15,16] when confronted with conflicts between their medical ethics and actual clinical difficulty.

In 2015, the Patient Right to Autonomy Act was enacted [17], pronouncing the expectations of respect for the patient’s medical autonomy and protection of the patient’s hospice rights, and putting forward the idea of “time-limited trials”. In particular, the aim of time-limited trials of withdrawing non-beneficial life-sustaining treatment (LST) before the end for critically and terminally ill patients with poor treatment responses is to ensure comfort in end-stage life and natural death [1,18,19].

As we all know, there is a gap between the ideal and reality, especially in terminal ICU patients’ end-of-life medical care. It is necessary to collect data to identify the actual situation and to be used as a reference for formulating acute and critical end-of-life care strategies and policy-making to promote the quality of end-of-life care for critically and terminally ill patients. 

### Objective

This study aims to conduct objective, non-interventional observation, and analysis through a two-year data collection and analysis study of ICU deceased to investigate the ratio of DNRs signed and the effect of DNR orders on the withholding of resuscitation practices and the withdrawal of life-sustaining treatments in terminally ill ICU patients with full medical coverage.

## 2. Methods

### 2.1. Design

This study was a retrospective study using the electronic medical records of terminally ill patients who received intensive care in the ICU before they died or had a predeath discharge between 1 January 2016 and 31 December 2017.

### 2.2. Sample and Setting

In this study’s institution, the ICU has 42 beds for critically ill adults, including 21 medical beds and 21 surgical beds, and belongs to a tertiary hospital in Taipei, Taiwan. In this study, the sampling process to control possible bias in the severity of the patients’ conditions was as follows. First, we identified a patient list that only counted each patient’s last admission to the ICU within the two years using the IntelliSpace Critical Care and Anesthesia System (ICCA), which the Department of Critical Care Medicine conducted in partnership with Philips. The entire patient list was 3636 people. Excluding 174 people who had been repeatedly admitted to the ICU, the remaining number was 3462 patients. Second, we excluded 1804 post-surgery patients and 198 patients after cardiac catheterisation. Third, we excluded 1074 patients who recovered and were discharged to the general ward. Finally, a total of 386 deceased cases were determined.

### 2.3. Ethical Considerations

The “Institutional Review Board” (IRB No.: 2019-04-009CC) and the “Human Research Protection Center” (HRPC No.: 1085000302) of the hospital approved this study before we conducted the data collection.

### 2.4. Instruments

The research used a self-developed recording form. The items were developed based on the study hypothesis and the electronic medical record of the hospital information system. The recording form contained 28 items (basic information: 7 items; disease information: 6 items; critical care: 3 items; DNR signed: 3 items; palliative consultations & care: 4 items; CPR and First Aid during ICU: 2 items; ICU discharge date & death place: 3 items) and was tested by five experts, resulting in a content validity index (CVI) of 0.79. The content items included the critically ill patients’ demographics and each patient’s worst physical condition within 24 h of being admitted to the ICU, which was determined by the Glasgow Coma Scale (GCS), Simplified Acute Physiology Score II (SAPS II) II, and Acute Physiology and Chronic Health Evaluation II (APACHE II), patient’s life-sustaining treatments, DNR orders, resuscitation practices, and hospice consultation.

### 2.5. Data Collection and Analysis

The first author collected the patients’ data, including admission information, disease information, critical care, DNR orders, palliative consultations and care, CPR and First Aid implementation, and ICU discharge date and death place from the ICU informatics system (IntelliSpace Critical Care and Anesthesia System; ICCA), and the corresponding author rechecked the patients’ data to ensure that no data were omitted. We used SPSS 24.0 as the statistical analysis software. In addition, the patients were divided into the DNR group and the non-DNR group, based on whether a DNR order was signed before death for the different analyses between the two groups. We applied the independent *t*-test and chi-square test to examine whether a DNR was signed with regard to withholding resuscitation practices and withdrawing life-sustaining treatments. This was carried out to test the study hypothesis that a DNR would positively affect the withholding of resuscitation practices and the withdrawing of life-sustaining treatments before death.

## 3. Results

We identified and collected the data of 386 terminally ill ICU patients who died in the ICU or who were predeath discharged to be included in this study. Of the study patients, 82.6% (319/386) had signed DNR orders and 17.4% (67/386) had not.

### 3.1. Patients’ Characteristics with Regard to Signing or Non-Signing of a DNR

In this study, 294 (76.2%) died in the ICU, and 92 (23.8%) were predeath discharged. Among the study patients, 319 (82.6%) signed DNR orders and 67 (17.4%) did not. Their average age was 65.62 years. Most were male, married, lived with their families, unemployed, and had religious beliefs of Buddhism or Taoism. The patient’s disease diagnosis specialties were haematology and oncology, gastroenterology, and infectious diseases, listed in descending order (Table 1). Ventilation care was the main reason for the patients’ admissions to the intensive care unit, followed by severe septic shock and massive bleeding. The average worst physical condition values in the first 24 h of ICU admission were an APACHE II score of 29.9, a SAPS II score of 73, and a GCS score of 7.8 (Table 2).

Compared to the non-DNR group, the DNR group was older and was less likely to be married (Table 1). In addition, for the worst physical conditions in the first 24 h of ICU admission, the GCS scores were similar. However, the DNR group’s SAPS II and APACHE II scores were significantly higher. There was no difference in the main reasons for admission to the ICU between the two groups (Table 2) and no difference in the times of ICU admission, hospitalisation, or the number of emergency room visits in one year between the two groups. Nevertheless, the DNR group had a significantly more extended stay in the ICU than the non-DNR group (Table 1). However, the proportion of patients who had completed a palliative care consultation before death in the DNR group was significantly higher than that in the non-DNR group (Table 1).

### 3.2. Influence of DNR Signing and Timing on Predeath Resuscitation Practice

The percentage of patients who had signed a DNR before admission to the ICU was 8.2% (26/319), while 32.9% (105/319) signed within 24 h of ICU admission, and 58.9% signed (188/319) after 48 h of ICU admission. This study identified the resuscitation practices in the ICU; these were conducted when there was cardiac arrest. As expected, compared to the non-DNR group, the DNR group had a significantly lower rate of chest compressions (4.7% vs. 41.8%), electric shocks (3.8% vs. 11.9%), and cardiotonic drug injections (26% vs. 53.7%). However, there was no significant difference in the percentage of patients having mechanical ventilation via an endotracheal tube (ET) between the two groups. Incredibly, it was also found that 75 DNR patients were treated with resuscitation practices after signing the DNR; furthermore, 73 DNR patients received cardiotonic drug injections (Table 3).

### 3.3. Signing the DNR for Withdrawing Life-Sustaining Treatments

Most patients were admitted to the ICU after being intubated or had an endotracheal tube inserted immediately after being admitted to the ICU. Therefore, this study supports using ET as a life-sustaining treatment to help the breathing of terminally ill ICU patients after resuscitation. The study results showed that the study patients had received a variety of life-sustaining medical treatments before they died, including ET (94%), antibiotics (96.4%), vasopressors (89.1%), blood transfusions (72.8%), and continuous venovenous haemofiltration (CVVH; 32.6%) (Table 4). A total of 71.7% of the patients had a central venous catheter (CVC) placed, and 95.1% had intra-arterial catheterisation (A-lines) for drug administration or vital sign monitoring (Table 2).

There was no significant difference in sedatives, muscle relaxants (Table 2), ECMO, or IABP (Table 4) between the non-DNR group and the DNR group. However, the DNR group had a significantly higher proportion of patients who received analgesics, antibiotics (Table 2), and CVVH than the non-DNR group (Table 4). However, only 5.4% of the patients withdrew from one or more life-sustaining treatments, which were as follows: twenty patients were removed from ET, four patients were withdrawn from vasopressors, two patients were withdrawn from CVVH, and one patient was withdrawn from IABP (Table 4).

## 4. Discussion

### 4.1. DNR Signing Ratio and Timing as an Ethical Dilemma

The study showed that not all patients (82.6%) signed the DNR order before death. The percentage of DNR signing was higher than that in previous studies ([11,20,21], which reported that less than 40% of patients had signed DNR orders before death. However, the signed DNR ratio was higher than the study result in 2010 under two similar ICU medical systems [11]. Taiwan has had palliative care legislation in the form of the Hospice Palliative Care Act since 2000 [10] and the Patient Right to Autonomy Act since 2016 [17]. This may be due to the effectiveness of palliative care legislation and the promotion of education to increase the signing of DNR to reject non-beneficial CPR for terminally ill ICU patients in order to reduce the suffering of patients before death. However, we noted that the proportion of DNRs signed before ICU admission was less than 10%, while about 1/3 were signed within 24 h of ICU admission, and more than half after 48 h of ICU admission. The study result was similar to the finding of Chang, et al. [22] that about 1/4 of ICU patients signed a DNR order within 24 h of ICU admission. One study indicated that compared to late DNR patients (after 48 h of ICU admission), the early DNR (within 48 h of ICU admission) ICU patients had fewer non-beneficial treatments, less perceived suffering, greater dignity, and more peaceful dying [23]. In addition, there are some arguments that the clinical decision of the “time-limited trial (TLT)” may be unethical if a patient has a meagre chance of survival, and the intensive care treatment would be potentially associated with harming patients by providing intensive care treatment and a negative dying experience [18,24,25]. From the study results, we can see that clinic professionals may struggle with this clinical ethical dilemma of DNR signing and the timing of signing, which would affect terminal patients’ end-of-life quality.

### 4.2. Illness Severity Assessment Had a Positive Effect on DNR Signing

The study results showed that the non-DNR group were in better physical condition upon admission to the ICU than the DNR group. However, they had more significant and rapid physical deterioration, resulting in a relatively short time from ICU admission to death. Notably, most patients’ DNR orders had been signed by their spouses or families when the patients were unconscious or unable to express their wishes clearly and independently after ICU admission [23]. However, many patients’ families have difficulty making DNR decisions for their loved ones, and choose to continue life-sustaining treatment to prolong their life [5], but patients who receive a family palliative care consultation are more likely to sign a DNR consent [26].

Compared to the non-DNR group, the patients with DNR orders had higher SAPS II scores and APACHE II scores on day one in the ICU than the non-DNR group. This indicates that the illness severity assessment for the patients on day one in the ICU would positively affect the signing of the DNR orders. This result is similar to a study by Miller, et al. [27]. That study identified that critically ill patients with more severe conditions, such as higher SAPS II and APACHE II scores, would increase the likelihood of DNR discussion and decision-making. It showed that ICU staff could use two valid and reliable assessment tools, SAPS II and APACHE II, to predict the mortality risk of critically ill patients in the ICU, especially within the first 24 h of ICU admission [28,29,30], to facilitate DNR consultation. However, we observed that the GCS scores were not significantly different between the two groups. The reason may be that the GCS scale is only used to assess coma severity. However, ICU patients often cannot achieve the verbal component score. Moreover, the GCS score is only one parameter of the SAPS II [31].

### 4.3. Signing DNR Orders Effectively Reduces Resuscitation Practices before Death

This study result shows the significant effect of palliative and hospice care promotion. A DNR decision means refusing some or all resuscitation practices, such as mechanical ventilation via an endotracheal tube (ET), chest compressions, cardiotonic medications, or electric shock. However, most of the patients in this study had received ET while being admitted to the ICU. Therefore, we examined the effects of signing a DNR and the cardiac resuscitation practices executed before death. The findings in this study are similar to the previous research, which found that the DNR order does reduce a patient’s chances of undergoing cardiac resuscitation when the patient has cardiac arrest as expected before death [11]. However, we observed that many terminal patients signed the DNR only after being admitted to the ICU because they might not have received candid information about their illness and advance care planning (ACP) before being admitted ICU. In all, 48.7% (188/386) signed a DNR order 48 h after ICU admission. This indicated that patients’ families sign the DNR orders for patients only after all treatments fail, and death is inevitable. [23,32,33]. Unfortunately, some patients had received cardiac resuscitation before the end, even after signing DNR orders. Approximately one-quarter of the DNR patients were treated with a cardiotonic drug injection when they were dying. The patients’ families may have thought that cardiotonic drug injections were less invasive and caused the terminal patients less suffering than chest compressions and electric shocks. ICU physicians accepted the patients’ families’ requirements and provided medical resuscitation treatments. Meanwhile, the medical professionals did not give up any chances for prolonging the patient’s survival [34]. The DNR order appears beneficial in limiting futile cardiac resuscitation practices to ensure terminally ill patients a peaceful natural death [24,26].

### 4.4. DNR Signing Did Not Facilitate the Withdrawal of Life-Sustaining Treatments

This study investigated whether signing DNR orders positively impacted the withdrawal of non-beneficial life-sustaining treatments for terminally ill ICU patients. In the study, antibiotics, vasopressors, ET, CVVH, and blood transfusions were often used as life-sustaining medical treatments for patients in the ICU. The result was the opposite of the assumption; signing a DNR order had no significant effect on the withdrawing of life-sustaining therapies, such as ET, vasopressors, antibiotics, and blood transfusions. This study shows that only 20 patients (5.2%) had their ET removed; the number was much smaller than expected. Because most patients continue to have an indwelling ET tube until death, clinicians have often used sedatives, muscle relaxants, and analgesics as comfort measures [35,36] to reduce the patient’s impedance of ET treatments [1,23,35]. This finding is similar to previous studies conducted in Taiwan [5,34]. The conclusions of this study were also identical to results in the survey, which showed that 80% of the patients who had signed DNR decisions still had life-sustaining treatments without reduction or withdrawal. However, life-sustaining treatments might prolong the patient’s suffering until the end of life, and could result in a higher bed occupancy rate and the consumption of medical resources in the ICU, especially in a country with national health insurance. Time-limited trials for critically and terminally ill patients have emerged in response to the increasing demand for intensive care. These would benefit critically ill patients with a low survival chance and be a solution to the clinical uncertainty inherent in critical care decision-making [18,21,24,25,37].

However, in the study, we observed that more than 90% of the patients in both groups received antibiotics, and the DNR group had a higher proportion of antibiotic treatment than the non-DNR group. This might be due to the DNR group having worse disease severity, such as SAPS II and APACHE II scores on ICU day one, than the non-DNR group [29]; another reason might be that the signing of the DNR orders was too late. We observed that less than half (131/319) of the patients signed a DNR before ICU admission or within 24 h of ICU admission in this study; therefore, more than half of ICU terminal patients would not have received a palliative care consultation, thus increasing the difficulty of withdrawal of life-sustaining therapies in medical ICUs. The Patient Right to Autonomy Act has been enforced in Taiwan since 2016 [17]. It proposes an option of a time-limited trial (TLT) to address the clinical dilemma of “rescue to the end” and improve the probability of withdrawal from the non-beneficial life-sustaining treatments in the end-of-life period. As one previous study recommended, we propose providing advance care planning (ACP) to help all patients be aware of the option of giving prior written and signed statements that express their medical decisions during the terminal stage [12]. For critically ill patients, we propose providing early DNR and TLT discussions while they are being admitted to the ICU. However, these recommendations need research to prove the effects of withholding futile resuscitation practices, especially cardiotonic drug injection, and of withdrawing non-beneficial of life-sustaining treatments of the ICU patients. This may prevent over-medicalisation and the prolongation of suffering at the end of life.

### 4.5. Limitations of the Study

This study was a retrospective study using two years of electronic medical records of ICU patients from a medical centre. Therefore, the results can only be extrapolated to intensive care units; this study has a limited application in general wards or hospice settings. Implementing the Patient Right to Autonomy Act may also affect this study.

### 4.6. Suggestions

Further studies are needed to investigate the benefit of DNR signing in the withholding of futile resuscitation practices, and to identify the practical influence of promoting “time-limited trials” on the withdrawing of non-beneficial life-sustaining treatments at the end of life after implementing the Patient Right to Autonomy Act. In addition, such research may also help prevent professionals from experiencing the moral distress of providing futile care to prolong death and unnecessary tests and treatments. It may also reduce the medical expenses of ineffective resuscitation practices and non-beneficial life-sustaining treatments.

## 5. Conclusions

Under the national health insurance system, the rights and interests of patients in intensive care are guaranteed, and the cost of intensive care for patients is paid in full. However, intensive care is not a panacea; patients may also deteriorate and enter the end stage of their lives while receiving life-sustaining medical care. This study found that most terminally ill ICU patients (82.6%) had signed DNR orders before death, and signing a DNR order was likely to lead to the withholding of resuscitation practice. However, 26% of patients received a cardiotonic drug injection after signing their DNR. In addition, the DNR order appeared less effective in terms of the withdrawing of life-sustaining treatment for ICU terminally ill patients at the end of life. In addition, this study also showed that patients with DNR orders had more probability of receiving palliative care consultations. However, we observed that the DNR group was in poorer physical condition (higher SAPS II score and APACHE II score) within 24 h of ICU admission, suggesting that ICU clinical professionals, while assessing critically ill patients for deteriorating health, would urge disclosure and hospice counselling and encourage DNR signing. The patients with DNR orders may have more likelihood of being spared non-beneficial resuscitation practices. Unfortunately, the study found that DNR signing did not help prompt withdrawal of existing non-beneficial life-sustaining treatments. It seems that DNR orders could not prevent the dilemma of withdrawing life-sustaining treatments for terminally ill ICU patients. 

In the future, studies should focus on how to urge discussion and decision-making on the effects of withholding futile resuscitation practices, especially cardiotonic drug injection, and withdrawing non-beneficial life-sustaining treatments of ICU patients to avoid over-medicalisation and to prevent the prolongation of suffering at the end. This kind of study will affect whether critically or terminally ill ICU patients can experience better end-stage quality of life and natural peaceful death.

### Application in Clinical Practice

This study’s findings would be an essential reference to help in understanding the clinical situation and in considering the importance of early assessment of the patient’s physical condition using practical tools to enable early prediction of patient health outcomes. In addition, ICU medical professionals should put forward hospice consultations to encourage patients to undertake early DNR signing to refuse resuscitation practices, and to facilitate decision-making with regard to “time-limited trials” of withdrawing non-beneficial life-sustaining treatments. This would help patients have a more comfortable and natural death.

## 6. Implications for Clinical Practice

(1)Learn about the current state of life-sustaining therapies in the ICU under the national health insurance support.(2)Facilitate the patient’s understanding of the benefits of natural death after signing a DNR order in the ICU.(3)Facilitate learning with regard to the low withdrawal rate of non-beneficial life-sustaining treatments in the ICU.(4)Provide a reference for considering the timing of DNR decision-making and the discussion of “time-limited trials” for withdrawal from non-beneficial life-sustaining treatments.

## Figures and Tables

**Table 1 ijerph-19-09521-t001:** Demographic parameters and comparison between the non-DNR and DNR groups.

Variable	Total *n* = 386	Non-DNR *n* = 67	DNR *n* = 319	*p*
Age	65.6 ± 17.8	63.1 ± 18.8	66.2 ± 17.5	0.418
Number of ICU admissions in one year	1.13 ± 0.49	1.15 ± 0.66	1.13 ± 0.44	0.788
Number of hospitalisations in one year	3.02 ± 3.32	3.34 ± 3.63	2.95 ± 3.25	0.41
Number of emergency room visits in one year	2.01 ± 2.39	1.76 ± 1.84	2.07 ± 2.49	0.344
Length of the last stay in ICU (days)	7.2 ± 9.16	4.79 ± 7.85	7.7 ± 9.34	0.018 *
Sex				0.266
Female	144 (37.3%)	29 (20.1%)	115 (79.9%)	
Male	242 (62.7%)	38 (15.7%)	204 (84.3%)	
Marital status				0.016 *
Married	249 (64.5%)	47 (70.1%)	202 (63.3%)	
Unmarried	56 (14.5%)	14 (20.9%)	42 (13.2%)	
Widowed/Divorced	81 (21%)	6 (9%)	75 (23.5%)	
Employed				0.315
Yes	135 (34.9%)	27 (40.3%)	108 (33.9%)	
No	251 (65%)	40 (59.7%)	211 (66.1%)	
Religion				0.781
None	151 (39.1%)	23 (33.4%)	128 (40.1%)	
Buddhism	139 (36%)	26 (39.4%)	113 (35.4%)	
Taoism	74 (19.2%)	16 (24.2%)	58 (18.2%)	
Others	22 (5.7%)	2 (3%)	20 (6.3%)	
Living style				0.251
Living with family	343 (88.9%)	60 (89.6%)	283 (88.7%)	
Living alone	23 (6%)	2 (3%)	21 (6.6%)	
Living in a nursing home	20 (5.2%)	5 (7.4%)	15 (4.7%)	
Specialty				0.231
Haematology and Oncology	109 (28.2%)	21 (31.3%)	88 (27.6%)	
Gastroenterology	79 (20.5%)	10 (14.9%)	69 (21.6%)	
Infectious diseases	63 (16.3%)	15 (22.4%)	48 (15%)	
Nephrology	37 (9.6%)	3 (4.5%)	34 (10.6%)	
Others	98 (25.4%)	18 (26.9%)	80 (25.1%)	
Palliative care consultation	101 (26.2%)	3 (4.5%)	98 (30.7%)	<0.001 ***
Death place				0.057
Died in ICU	294 (76.2%)	45 (67.2%)	249 (78.1%)	
Predeath discharged	92 (23.8%)	22 (32.8%)	70 (21.9%)	

Note: Significant difference between the two groups using an independent two-sample *t*-test/a chi-squared test, * *p* < 0.05, *** *p* < 0.001.

**Table 2 ijerph-19-09521-t002:** Reasons for ICU admission, physical condition, comfort line, and comfort care comparison between the non-DNR and DNR groups.

Items	Total *n* = 386	Non-DNR *n* = 67	DNR *n* = 319	*p*
Worst physical conditions within 24 h of ICU admission				
GCS score	7.8 ± 3.9	7.5 ± 4.1	7.9 ± 3.9	0.425
SAPS II score	73 ± 26	66.6 ± 28.2	74.4 ± 25.4	0.028 *
APACHE II score	29.9 ± 9.2	27.5 ± 10.6	30.4 ± 8.8	0.037 *
Care needs for ICU admission				0.996
Respiratory failure	266 (68.9%)	47 (70.1%)	219 (68.7%)	
Severe septic shock	49 (12.7%)	9 (13.4%)	40 (12.5%)	
Massive haemorrhage	14 (3.6%)	2 (3%)	12 (3.8%)	
Post-resuscitation	30 (7.8%)	5 (7.5%)	25 (7.8%)	
Diabetic ketoacidosis	1 (0.3%)	0 (0%)	1 (0.3%)	
Severe heart failure and acute pulmonary oedema	8 (2.1%)	1 (1.5%)	7 (2.2%)	
Drug toxicity with organ failure	3 (0.8%)	0	3 (0.9%)	
Electrolyte imbalance of body fluids associated with renal failure	4 (1%)	1 (1.5%)	3 (0.9%)	
Others	11 (2.8%)	2 (3%)	9 (2.8%)	
Comfort care				
Sedatives	166 (43%)	25 (37.3%)	141 (44.2%)	0.301
Muscle relaxants	68 (17.6%)	9 (13.4%)	59 (18.5%)	0.323
Analgesics	173 (44.8%)	20 (29.9%)	153 (48%)	0.007 **
Vascular line				
An intra-arterial catheter (A-line)	367 (95.1%)	63 (94%)	304 (95.3%)	0.663
Central venous catheter (CVC)	277 (71.7%)	43 (64.2%)	234 (73.4%)	0.129

Note: Significant difference between the two groups using an independent two-sample *t*-test/a chi-squared test, * *p* < 0.05, ** *p* < 0.01; GCS: Glasgow Coma Scale; SAPS II: Simplified Acute Physiology Score II; APACHE II: Acute Physiology and Chronic Health Evaluation II.

**Table 3 ijerph-19-09521-t003:** Influence of DNR signing and timing on predeath resuscitation practice.

Item	Non-DNR *n* = 67	DNR *n* = 319	*p*	Time of DNR Signed			Resuscitations after DNR *n* = 75
				Before ICU Admission *n* = 26	Within 24 h of ICU Admission *n* = 105	48 h after ICU Admission *n* = 188	
Chest compressions	28 (41.8%)	15 (4.7%)	<0.001 ***	1 (3.8%)	6 (5.7%)	8 (4.3%)	4 (5.3%)
Electric shock	8 (11.9%)	4 (1.3%)	<0.001 ***	1 (3.8%)	2 (1.9%)	1 (0.5%)	1 (1.3%)
Cardiotonic drugs injection	36 (53.7%)	83 (26%)	<0.001 ***	9 (34.6%)	37 (35.2%)	37 (19.7%)	73 (97.3%)

Note: Significant difference between the two groups using a chi-squared test, *** *p* < 0.001.

**Table 4 ijerph-19-09521-t004:** Life-sustaining treatments were executed and withdrawn between the non-DNR group and the DNR group.

	Total *n* = 386	Non-DNR *n* = 67	DNR *n* = 319	*p*	Withdrawal after DNR *n* = 319
Endotracheal tube & ventilator (ET)	363 (94%)	64 (95.5%)	299 (93.7%)	0.355	20 (6.27%)
Vasopressors	344 (89.1%)	64 (95.5%)	280 (87.8%)	0.064	4 (1.25%)
ECMO	18 (4.7%)	4 (6%)	14 (4.4%)	0.577	2 (0.63%)
Continuous venovenous haemofiltration (CVVH)	126 (32.6%)	15 (22.4%)	111 (34.8%)	0.049 *	2 (0.63%)
Haemodialysis	37 (9.6%)	5 (7.5%)	32 (10%)	0.516	0
Antibiotics	372 (96.4%)	61 (91%)	311 (97.5%)	0.01 *	0
Blood transfusion	281 (72.8%)	44 (65.7%)	237 (74.3%)	0.149	0
Intra-aortic balloon pump (IABP)	4 (1%)	2 (3%)	2 (0.6%)	0.083	1 (0.31%)

Note: Significant difference between the two groups using a chi-squared test, * *p* < 0.05; 21 out of 389 patients who died were withdrawn from life-sustaining treatments. In detail, 16 patients were removed from the ventilator; two patients were removed from the ventilator and stopped taking blood pressure medicines; one patient was removed from the ventilator and stopped blood pressure medicines, ECMO, IABP, and CVVH; one patient was withdrawn from ECMO and stopped blood pressure medicine; and one patient had the respirator removed and CVVH.

## Data Availability

Data are not permitted to be publicly available to preserve the confidentiality of participants; however, they are available from the corresponding author upon specific reasonable request.

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
