# Peer review of "The Impact of Signing Do-Not-Resuscitate Orders on the Use of Non-Beneficial Life-Sustaining Treatments for Intensive Care Unit Patients: A Retrospective Study"

_ijerph, 2022, doi:10.3390/ijerph19159521_

Round 1
Reviewer 1 Report
Thank you for the opportunity to read your manuscript. While it provides a very interesting reflection of what happens in one institution in Taipei, I am afraid I have not suggested that it is published, because I do not think it is of international relevance, nor does it reflect recent research in the field. There has been an awareness that DNR can sometimes be conflated with other treatments being withheld, and a move towards making sure - especially on ICU - that resuscitation decisions are always contextualised within overall goals of care.
I wonder whether your paper might be of interest to national readership, and if so, I suggest you redraft it with some context of the law around DNR decisions ( who can make them, if they are legally bindings etc) and perhaps a comparison with other countries?
It is clear that you have done hard work on the data collection and analysis, which is very good.
Wishing you luck!
Reviewer 2 Report
The abstract describes the study's objective precisely, and this version of the objective should be considered used throughout the manuscript.
The clarity of the conclusions could be improved, i.e., according to the objective.
Abbreviations should not be used alone before they are explained, including in the heading and the abstract.
The DNR is referred to repeatedly and should be presented in extenso, as this is important for the practical application and interpretation of the DNR. The authors should present the legal status of the DNR. Concerning respecting the patient's autonomy, how was the potential need for a DNR declaration presented to the patients and family members, including a description of the patient's level of consciousness related to, e.g. disease and medication? Who is involved in the team and responsible for the DNR implementation?
Ethically, one should always be clear how a DNR procedure is far from being any kind of euthanasia.
Regarding the hospital's quality assurance management, including the documentation of recordings routines: These systems should be described, including improvement proposals.
The conclusion should be more clearly related to the objective.
The research design seems ok. I have not controlled the statistics involved.
Reviewer 3 Report
This fascinating paper explores a relevant theme for patients, families, and health professionals "The Impact of Signing DNR Orders on the Use of Non-Beneficial Life-Sustaining Treatments for Intensive Care Unit Patients: A Retrospective Study."
It is not easy to decide whether or not to DNR. This paper is written clearly and focuses the reader's attention while exploring the main aspects that are the focus of complex decision-making since they fall in the ethics of promoting or prolonging the patient's life.
Although the simplicity of the statistical methodology and the exploratory and case study design, the paper may trigger the interest in discussing this theme more deeply. The introduction is well-grounded, but I would suggest the authors explain the process that led to the signature of DNR. That aspect is essential to understanding the paper's relevance. Not every country adopts the same procedure, so the authors should clarify how and why this study must be read and spread worldwide. The methodology is clear and replicable. The instruments used could be explained in more detail. The ethical procedures were accounted for. The results and the discussion are coherent with the aim of the study. The limitations are adequately identified, but I would like the authors to explain why they did not consider other health units – it could be a way to confirm the results and contribute to a higher balance between the groups that are the target of comparison. The difference between groups is significant, which also can constitute a limitation.
